# Dual-Band Frequency Selective Surface with Different Polarization Selectivity for Wireless Communication Application

**DOI:** 10.3390/s23094264

**Published:** 2023-04-25

**Authors:** Tao Qin, Chenlu Huang, Yang Cai, Xianqi Lin

**Affiliations:** 1School of Electronic Science and Engineering, University of Electronic Science and Technology of China, Chengdu 611731, China; 2Yangtze Delta Region Institute (Huzhou), University of Electronic Science and Technology of China, Huzhou 313001, China; 3Department of Electrical and Information Technology, Lund University, 22100 Lund, Sweden; 4Department of Electronic Systems, Aalborg University, 9220 Aalborg, Denmark

**Keywords:** dual-band, frequency selective surface, polarization selective selectivity, wireless communication, duplex system, meta-surface

## Abstract

This article proposes a dual-band, frequency- and polarization-selective surface. Multiple resonant modes are introduced using the U-shaped resonator with a ground via to achieve dual-band responses and polarization selectivity. Two symmetrically grounded U-shaped resonators are coupled through electrically coupled apertures in a common ground, resulting in a passband with two transmission zeros per polarization. A general design flowchart and additional examples at the S, X, and K-bands are presented as well. A prototype at X-band is analyzed, fabricated, and measured, showing the passband center frequencies of 9.68 GHz and 10.73 GHz, factional bandwidths of 3.45% and 3.48%, and insertion losses of 0.9 dB and 1.1 dB, respectively. Due to the high selectivity, small frequency ratio, low profile, and stable performance under oblique incidence, the proposed designs have application potential in wireless communication systems.

## 1. Introduction

Modern wireless communication systems have adopted frequency duplex (FD) and polarization duplex (PD) as solutions for simultaneous transmission and reception to meet the growing demand for higher transmission data rates [1,2,3,4]. These techniques are used in numerous applications, including satellite communication [5,6], mobile networks [7,8], vehicular communication [9], and radio-frequency identification (RFID) systems [10,11], resulting in an increase in channel capacity and a decrease in data collision rates.

In recent years, as a result of advances in electromagnetic meta-surface research and development, frequency selective surfaces (FSS) and polarization selective surfaces (PSS) have emerged as promising duplex technologies in wireless systems, covering areas such as antenna design [12,13], analogue circuit absorption [14,15], polarization diversity [16], spatial filtering [17], and beam-switching [18,19]. Dual-band FSSs can be designed by cascading multi-layer structures [20,21,22], employing multi-resonant FSS units [23], designing fractal elements [24,25,26], and adopting the perturbation of a single-layer FSS [27]. In Ref. [22], arrays of double square loops and wire grids are cascaded to create an X- and Ka-band transmission-type dual-band FSS. An ultra-thin single-layer multi-resonant dual-band FSS is proposed in Ref. [27], which consists of a square patch with eight L-shaped arms. The L-shaped arm’s vertical and horizontal portions produce a stopband, respectively. In addition, the emergence of three-dimensional FSS offers a new design method for dual-band FSSs [28,29]. On the other hand, multi-layer film stacks, grating design, and cascading planar resonators are the primary design techniques in the electromagnetic field for PSSs [30,31,32,33,34]. Ref. [31] describes a simple U-shaped metallic structure that generates a first-order passband for transverse electric (TE) and transverse magnetic (TM) incident waves, respectively. Due to the use of a single resonating mode and a single substrate layer, however, the performance of selectivity and polarization isolation is limited. In Ref. [32], the polarization grid in a folded reflectarray antenna is replaced with a FSS that has two elliptical patches separated by ground, and an elliptical slot. This FSS is designed to have different polarization selectivity in each of the two frequency bands that the antenna is meant to operate in. This modification allows the reflectarray to support dual-polarization and operate in both frequency bands. In Ref. [33], a novel PSS composed of three metallic layers is proposed. Based on the resonating mechanism of the Jerusalem cross, this design offers two second-order passbands under the TE wave at 6.5∼8.07 GHz and 25.42∼28.46 GHz, respectively. In the frequency range of 9.94 GHz to 12.88 GHz, the best reflectivity for TM wave is approximately −40 dB. Similarly, in Ref. [34], a PSS composed of three metallic layers, each with a square loop and circular patch with a twisted slot, generates two orthogonal linearly polarized waves in the lower (7.63∼8.97 GHz) and upper (13.26∼13.87 GHz) frequency bands. Still, there are some design disadvantages and application limitations in the works above:Current designs feature a frequency ratio that is notably high, making the design of the feeding networks and signal processing modules in frequency division duplex (FDD) systems more difficult and costly, particularly in satellite communication and base station applications. In addition, despite the widespread use of circuit analysis in the literature, the relationship between circuit components, physical structure, achievable performance, and the general design process applicable to different frequency bands has not been thoroughly explored;There are few studies on the in-band flatness and out-of-band rejection in the current research on polarization and frequency selective surfaces of duplex systems. It is rarely addressed how to introduce one or more transmission zeros (TZs) at the cross-polarized frequency to improve polarization isolation, particularly when the two orthogonal polarizations operate at close frequencies.

This article describes a dual-band frequency selective surface with polarization selectivity, whose two passbands transmit orthogonally polarized incident waves, respectively. The proposed design consists of two identical U-shaped resonators, a common ground with two coupling apertures, and two dielectric substrates to isolate the U-shaped resonators from the ground. Importantly, each U-shaped resonator is connected to the ground via a metallic via, allowing for multiple modes of resonance with different polarizations. Two transmission zeros are introduced at the lower and upper band edges of the two passbands to significantly improve the frequency selectivity and cross-polarization suppression. Finally, a prototype with an operating frequency of approximately 10 GHz is fabricated and tested to validate the design methods discussed. The following is a summary of this article’s main contributions:By grounding the U-shaped resonant structure, a small frequency ratio between the two orthogonal polarizations can be achieved. By adjusting the substrate thickness, the radius of the grounded via, and the dimensions of the U-shaped resonator, the frequency ratio achievable at X-band ranges between 1.05 and 1.4;A general framework is provided. In addition, full-wave simulations and equivalent circuits are analyzed and were found to agree well, which would be of great assistance in comprehending the mechanism of our structure and optimizing its parameters during design. Followed by step-by-step explanations and design examples for the S-, X-, and K-bands, the most significant contribution is a summary of the general design flow diagram. The proposed designs have significant application potential in modern wireless communication and multiple-input, multiple-output (MIMO) systems due to their high selectivity, small frequency ratio, low profile, and stable performance under oblique incidence.

## 2. Operation Principles and Analysis Methods

The detailed operating principle of the fundamental U-shaped resonator is discussed in this section, followed by parameter studies. Next, a multi-layer meta-surface composed of two identical U-shaped resonators, a common ground, two coupling apertures, and two dielectric substrates is provided. The equivalent circuit model of the proposed meta-surface is then analyzed and validated via comparison with the results of the full-wave simulations. Finally, the response under oblique incidence is investigated, and a comprehensive design flowchart is summarized.

### 2.1. Dual Modes U-Shaped Resonator with One Grounded Metallic Via

Figure 1a depicts the fundamental U-shaped resonator structure used in this article. The full-wave analysis of this structure is performed using the High-Frequency Structure Simulator (HFSS). Using the periodic boundary condition (PBC) in HFSS, the electric field of the TM polarization incident Floquet mode is along the positive *x*-axis, whereas the electric field of the TE polarization is along the positive *y*-axis. Due to its wide frequency response, the U-shaped resonant ring composed of a microstrip line structure has been widely used in antenna and feeding structure design over the past few decades [35,36,37,38]. Nonetheless, its application in FSS and PSS has received limited consideration [31,39,40]. Figure 1b demonstrates that the metallic portion of the U-shaped ring consists of two open-ended arms and a stub with a long edge along the *y*-axis. The specific geometric dimensions of the adopted U-shaped resonator are detailed in Table 1. Our designs employ Taconic RF-35 substrates with a dielectric constant of 3.50 and a loss tangent of 0.002.

Figure 2a illustrates the relationship between the reflection coefficient of a U-shaped resonator without a grounded via and the operating frequency for TE and TM polarized incident waves. The metallic surface current distributions under different incident polarizations are depicted in Figure 2b,c, respectively. When the incident polarization is TM, the incident electric field is along the *x*-axis direction, which results in an even-mode current flowing along the stub and two anti-phase currents in the two arms. On the other hand, under TE polarization, the incident electric field is along the *y*-axis direction, which excites two in-phase currents in the two arms and an odd-mode distribution along the stub. Therefore, the transmission line model depicted in Figure 3 is applied to analyze the response of this structure. The characteristic impedance of the microstrip line, where the source line width is $l_{c}$ and *w*, is denoted as $Z_{stub}$ and $Z_{arm}$ [41,42], respectively. The propagation constants corresponding to these transmission lines are denoted as $\beta_{stub}$ and $\beta_{arm}$. A short transmission line with a length of $l_{open}$ is introduced as the equivalent compensation amount due to the open-ended condition [42]. The influence of bending and chamfering is disregarded here and can be corrected with minor optimization later on.

For the TM polarization, the symmetry plane $AA_{1}$ is the equivalent magnetic wall and the resonance condition can be expressed as follows [42]:(1)$$\frac{Z_{arm}}{\tan\left(\frac{l_{y}+l_{open}}{\beta_{arm}}\right)}-Z_{stub}\tan\left(\frac{l_{x}}{\beta_{stub}}\right)=0$$

As for the TE polarization, the symmetry plane $AA_{1}$ is the equivalent electric wall and the resonance condition is:(2)$$Z_{arm}\tan\left(\frac{l_{y}+l_{open}}{\beta_{arm}}\right)-Z_{stub}\tan\left(\frac{l_{x}}{\beta_{stub}}\right)=0$$

In the frequency range of 7 to 13 GHz, as depicted in Figure 2a, this structure exhibits a resonance characteristic at 10.2 GHz that is exclusive to the TM polarization, as confirmed by Equation (Equation 1). According to Equation (Equation 2), the calculated resonance frequency for the TE polarization is approximately 19 GHz, resulting in a significantly smaller current amplitude at 10.2 GHz compared to the TM polarization.

Equations (Equation 1) and (Equation 2) illustrate that when the arm width *w* is close to the stub width $l_{c}$, the resonance frequency ratio between TE and TM polarization approaches 2:1. To meet the smaller frequency ratio requirements of FDD systems and to tune the resonance frequencies, a metallic grounded via with a radium of $r_{v}$ is positioned in the stub center. The via-hole grounds in the microstrip line can be represented by the inductance of a cylindrical conductor derived from the Maxwell equations [43,44,45]. Figure 4 illustrates the new structure simulation results. Under TM polarization, the influence of the introduced inductance on the resonant frequency is minimal because the symmetry plane functions as a magnetic equivalent boundary. Nonetheless, the resonant frequency for TE polarization decreases to 9.31 GHz when a significant odd-mode current is activated. In this case, the resonant frequency ratio is adjusted to approximately 1.11, which is close to the norm for duplex systems.

### 2.2. Parametric Study of the Grounded U-Shaped Resonator

The preceding section provides a general overview of the resonance conditions for TE and TM polarizations. This subsection will delve deeper into parameter studies. Notably, the substrate parameters, such as height *h* and permittivity value, are not included in the subsequent analyses.

$l_{x}$ and $l_{y}$: Lengths of the stub and arm have a strong effect on the resonant frequencies. Typically, when the widths of the stub and the arm are comparable, resonance occurs when the sum of the electrical lengths of the stub and the arm is approximately half a wavelength. The resonant frequencies are then estimated to be:
(3)$$f_{odd}\approx f_{even}\approx \frac{c}{2\times \left(2l_{x}+2l_{y}\right)\sqrt{\varepsilon_{\mathrm{eff}}}}\approx \frac{c}{4\left(l_{x}+l_{y}\right)\sqrt{\frac{\varepsilon_{r}+1}{2}}}$$
where, to simplify the calculation, the equivalent permittivity expression without regard to the strip width and substrate height, εeff=(εr+1)(εr+1)22, is used here to estimate the operating wavelength of the microstrip line. Equation (Equation 3) shows that when $l_{x}$ or $l_{y}$ increases, the resonant frequency will decrease. For further validation, the full-wave simulation results of $l_{x}$ and $l_{y}$ parameter sweeping are given in Figure 5a,b, respectively.$l_{c}$ and *w*: The width of the stub and arms primarily affects the characteristic impedance and propagation constant of their respective transmission lines, which have substantial effects on the resonance frequencies and input coupling bandwidth, i.e. the loaded quality factor. In addition, the ratio of $l_{c}$ to *w* influences the frequency ratio between TE and TM resonances. In general, as shown in Figure 5c, the loaded quality factor increases as the width increases. The closer the ratio of $l_{c}$ to *w* approaches 1, the closer the resonant frequencies of the two modes become.$r_{v}$: This parameter primarily affects the equivalent inductance introduced by the grounded metallic via, which in turn influences the tuning effect on the TE resonance frequency. As shown in Figure 5d, the impact of this parameter on TM resonance is minimal due to the presence of an equivalent magnetic wall. Consequently, the ratio of resonant frequency can also be tuned by adjusting the radium of the grounded via.

During our design, through the proper selection of parameters and modification of the dielectric substrate thickness and permittivity, resonant frequency ratios in the range of 1.04 to 2.2 can be achieved.

### 2.3. Frequency Selection and Cross-Polarization Suppression Introduced by Coupling Apertures

To meet the requirements for dual polarization isolation and signal bandwidth in wireless communication systems, the structure under discussion must incorporate multiple resonances and cross-coupling structures to achieve a flat transmission coefficient in the passband and multiple TZs outside of the passband. Figure 6 depicts the structure proposed in this paper with dual-frequency polarization selection characteristics. Two Taconic RF-35 substrates with a shared ground layer compose this structure. Each layer possesses a U-shaped, uniformly sized resonant ring that is grounded. On the ground layer, two identical coupling apertures are etched close to the open ends of the arms, where the electric field is at its strongest. In both the xoy and yoz planes the entire structure is symmetrical. The addition of the coupling aperture generates dual-port transmission characteristics and two additional out-of-band TZs. The physical parameters of this structure are listed in Table 2.

Table 3 presents the electric field and surface current distributions of the U-shaped resonator and ground under the TE wave incidence at 9.54 GHz and the TM wave incidence at 10.56 GHz, respectively. Under TE polarization, an odd-mode resonance is observed, whereas under TM polarization, an even-mode resonance is supported. The equivalent circuit model with a uniform topology, as shown in Figure 7, can therefore be utilized to analyze the design in both polarizations.

In Figure 7, all materiel-caused losses are disregarded, and the free space is represented by two infinite transmission lines with a characteristic impedance of $Z_{0}=377$ Ω. Each dielectric substrate is equivalent to a short transmission line with a characteristic impedance of $Z_{s}=Z_{0}/\sqrt{\varepsilon_{r}}$ and a length of $l_{s}=h$, where $\varepsilon_{r}=3.5$. The U-shaped resonator is replaced by the series connection of an inductance $L_{u}$ and two capacitors $C_{u1}$ and $C_{u2}$. The parallel combination of an inductance $L_{a}$ and a capacitance $C_{a}$ represents the equivalent elements of the two symmetrical apertures for coupling. The capacitance $C_{m}$ represents the electrical coupling between the two U-shaped resonators, whereas the coupling between the U-shaped resonator and the ground is relatively small and insignificant. In addition, metallic vias have the same inductance $L_{v}$ as discussed in Ref. [43]. The region of the ground with a strong current distribution is modeled as an inductor with a value of $L_{g}$.

Since the equivalent circuit is symmetrical at the $A_{1}A_{2}$ plane, even-odd-mode analysis can be utilized once more. Significantly, each short transmission line can be replaced by a series inductor with a value of $L_{t}=\mu_{0}\mu_{r}h$ and a shunt capacitor with a value of $C_{t}=\varepsilon_{0}\varepsilon_{r}h$, where $\mu_{0}$ and $\varepsilon_{0}$ are the permeability and the permittivity of the free space, respectively, and $\mu_{r}$ is the relative permeability of the substrate. For the even-mode resonance, an equivalent magnetic wall is inserted at the symmetry plane $A_{1}A_{2}$. The simplified equivalent circuit is shown in Figure 7b and the input admittance is expressed as:(4)$$Y_{even}=j\omega C_{t}+\frac{L_{v}}{j\omega M}+{\left\{{\left(\frac{2L_{g}}{j\omega M}\;+j\omega C_{u1}\right)}^{-1}+{\left[\frac{L_{t}}{j\omega M}+\frac{j\omega \left(C_{u2}-C_{m}\right)}{1-\omega^{2}L_{u}\left(C_{u2}-C_{m}\right)}\right]}^{-1}\right\}}^{-1}$$
where the ω is the angular frequency and variable M is given by:(5)$$M=2L_{t}L_{g}+2L_{v}L_{g}+L_{t}L_{v}$$

Similarly, the input admittance of the odd-mode resonance, as shown in Figure 7c, obtained by inserting an equivalent electrical wall, is expressed as:(6)$$Y_{odd}=j\omega C_{t}+\frac{1}{j\omega L_{t}}+{\left\{\frac{1}{j\omega C_{u1}}+{\left[\frac{1}{j\omega L_{v}}+\frac{j\omega \left(C_{u2}+C_{m}\right)}{1-\omega^{2}L_{u}\left(C_{u2}+C_{m}\right)}\right]}^{-1}\right\}}^{-1}$$

Then the S-parameters of this structure can be derived as [42]:(7)$$S_{11}=S_{22}=\frac{Y_{0}^{2}-Y_{even}Y_{odd}}{\left(Y_{0}+Y_{even}\right)\left(Y_{0}+Y_{odd}\right)}$$
(8)$$S_{21}=S_{12}=\frac{\left(Y_{even}-Y_{odd}\right)Y_{0}}{\left(Y_{0}+Y_{even}\right)\left(Y_{0}+Y_{odd}\right)}$$
where $Y_{0}=1/377$ $\Omega^{-1}$ is the characteristic admittance of the air-space port.

Figure 8 displays the results of the HFSS full-wave simulation and equivalent circuit calculations. With the exception of a minor disparity, the results are in good agreement. Table 4 provides the values for the synthesized circuit elements. The slight difference may be the result of ignoring the equivalent circuit of the U-shaped resonators’ coupling to the ground.

### 2.4. Sensitivity Analysis and General Design Flowchart

The impact of the geometric parameters of the FSS on the resonant frequencies, transmission zeros, and bandwidths is illustrated in Figure 9. Figure 9a demonstrates that when the sum of $l_{x}$ and $l_{y}$ is held constant, increasing the value of $l_{x}$ results in a reduction in the frequency ratio between the two passbands, which aligns with the previous analysis. As shown in Figure 9b, the size of the coupling apertures $a_{x}$ and $a_{y}$ primarily affects the coupling amount of the two U-shaped rings, i.e., the passband bandwidth and the TZ positions. To examine the design sensitivity to the incident angle, Figure 9c,d displays the transmission coefficients of the FSS under varying incident angles in the xoz and yoz planes, respectively. In the yoz plane, the transmission response remains stable for incident angles ranging from 0 to 50 degrees. When the incident vector is in the xoz plane, the TM incident wave activates the even mode, leading to dual-band behavior. Nevertheless, the in-band cross-polarization suppression in the lower passband still exceeds 13 dB even at an incident angle of 30 degrees.

Figure 10 provides general design guidelines for the proposed dual-frequency and polarization-selective meta-surfaces. Detailed design steps of the proposed X-band FSS are given here:

Step 1: The center frequencies of the dual frequencies are set to 9.7 GHz for TE and 10.7 GHz for TM, which corresponds to an approximate frequency ratio of 1.1. Considering that the adopted resonant cell is based on a microstrip U-ring structure, the achievable 3-dB fractional bandwidth (FBW) is between 3% and 7%; therefore, the target bandwidth requirement is set to 5%.

Step 2: Here, we utilize Taconic RF-35 substrates. The relative permittivity is set to 3.5, and the single-layer thickness is 0.508 mm. To reduce the errors introduced by finitely periodical boundaries and to improve the stability under oblique incidence, the recommended cell size is between one-third and one-half wavelength. Therefore, in this case, both $P_{x}$ and $P_{y}$ are set to 5 mm.

Step 3: Initialize $l_{c}$ and *w* to the same value, 0.8 mm. This value can be initially set to the width corresponding to a 50-ohm microstrip line and optimized for a range of approximately 20 to 120 ohms when input coupling is considered. To satisfy the processing conditions, the grounded via radius is set to 0.2 mm. It is initially assumed that the arm has the same length as the branch, that is, $l_{y}=2l_{x}$. According to Equation (Equation 3), the value of $l_{y}$ is approximately 3.33 mm. Then, minor optimization in HFSS is used to maintain the center frequencies at 9.7 GHz and 10.7 GHz, respectively. The optimized parameters, which are all close to their initial values, are $l_{y}=2l_{x}=3.04$ mm, w=0.9 mm, and $l_{c}=0.8$ mm.

Step 4: Build the whole meta-surface model shown in Figure 6 with the values obtained in step 3. The initial coupling aperture size is set to $a_{x}=2a_{y}=w$. Then, the element values in the equivalent circuit are fitted to the full wave simulation results. In practice, only the value of $C_{a}$, $L_{a}$, and $L_{g}$ must be fitted. The value of the remaining elements can be calculated from the transmission line models or resonance models. Due to the introduction of the cross-coupling, the self-resonant frequency of the resonator will be affected, and a small correction to $C_{u}$ and $L_{u}$ is required. During our design, this correction range is within 25%.

Step 5: Modify the simulation model based on the equivalent circuit obtained in step 4. Adjust the values of $l_{x}$ and $l_{y}$ for resonance tuning, as well as the values of ax and ay for the coupling amount, i.e., the values of $C_{a}$, $L_{a}$, and $L_{g}$. Numerical optimization is possible using Equations (Equation 7) and (Equation 8). Four or five iterations for each polarization will bring the simulation results closer to the desired value.

Step 6: Minor parameter optimization in HFSS, where the value ranges are all within 3%, is carried out to ensure good in-band flatness and the required bandwidth. The final physical dimensions and elements values have been listed in Table 2 and Table 4, respectively.

To further demonstrate this design method, additional simulation results for different operating frequency bands are presented in Figure 11. In both of these design examples, the topology structure shown in Figure 6 and the equivalent circuits shown in Figure 7 are still used, and the dimensions corresponding to these structures are listed in Table 5. For the S-band design shown in Figure 11a, the center frequencies are approximately 2.25 GHz and 2.55 GHz, while the polarization isolation is around 77 dB and 75 dB, respectively. The center frequencies and polarization isolation for the K-band design depicted in Figure 11b are approximately 18.7 GHz and 20.7 GHz, and 46 dB and 43 dB, respectively. Noting that the insertion losses in these designs at S-, X-, and K-bands are all less than 0.4 dB, the proposed polarization and frequency selective meta-surfaces are applicants for use in modern wireless communication systems.

## 3. Fabrication and Measurement Results

To evaluate the performance of the proposed dual-band PSS depicted in Figure 8, a prototype comprised of 36 × 36 unit cells is fabricated and tested. The thickness of the single copper layer is 0.018 mm and the overall dimensions of the prototype are 180 mm × 180 mm × 1.07 mm. Figure 12 illustrates the prototype and the measurement setup. These setups and detailed measurement procedures are nearly identical to those described in our earlier work [17]. As transmitting and receiving antennas, respectively, two standard gain horns are connected to a two-port vector network analyzer. The FSS prototype is placed between the two horn antennas and in a square aperture of the absorbing materials. Far-field conditions are ensured by the separation between the transmitting antenna and the prototype. To calibrate the measured results, the transmission coefficients without FSS under various incident angles are first measured.

Figure 13 presents the transmission coefficients measured in xoz and yoz planes. The center frequencies of the two measured passbands are 9.68 GHz and 10.73 GHz, with the worst insertion losses of 0.9 dB and 1.1 dB, respectively. The 3-dB FBW of each passband is around 3.5%, and the minimum and maximum in-band cross-polarization suppressions under normal incidence are 30.4 dB and 41.7 dB, respectively. Due to the TZs introduced on both sides of the passband, the out-of-band rejection reaches 20 dB for both polarizations. Importantly, the response of the prototype remains stable with the incident angle varying within the range of 0∼30 degrees in the xoz plane and 0∼50 degrees in the yoz plane. Note that the measured operating frequency, the insertion loss, and the depth of transmission zeros are all slightly higher than their simulated values. This discrepancy between simulation and measurement may be attributable to the substrate dielectric constant imprecision, fabrication error, and measurement error. Table 6 shows the comparison between the fabricated design and the dual-band PSS described in other published works. The structure proposed in this paper has a low profile and design difficulty, a small frequency ratio, high cross-polarization isolation, and out-of-band suppression in comparison to existing literature.

## 4. Conclusions

This article introduces a dual-band frequency selective surface that displays different polarization selectivity in two passbands. The two passbands utilize orthogonally polarized incident waves and each band edge has a transmission zero. Using accurate equivalent circuit modeling and even-odd mode analysis, the operating principle is discussed. The article also includes a general design flowchart and two additional designs. To verify its performance, an X-band prototype has been fabricated and measured, and the results are in good accordance with the simulation results. A prototype at X-band is analyzed, fabricated and measured, showing the passband center frequencies of 9.68 GHz and 10.73 GHz, factional bandwidths of 3.45% and 3.48%, and insertion losses of 0.9 dB and 1.1 dB, respectively. The structure proposed in this paper has a low profile and design difficulty, a small frequency ratio, high cross-polarization isolation, and out-of-band suppression in comparison to existing literature. Therefore, this design has the potential as a device for contemporary wireless communication systems.

Our future work will aim to design frequency- and polarization-selective surfaces with higher order and wider bandwidth by introducing additional structures, i.e., symmetrical microstrip branches and grounded vias. Moreover, attempts will be made to realize meta-surface designs with linearly polarized rotation properties by utilizing double-layered asymmetric structures.

## Figures and Tables

**Figure 1 sensors-23-04264-f001:**
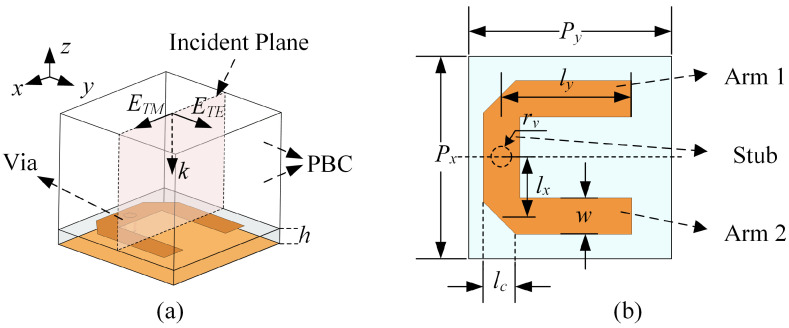
Structure of the U-shaped resonator used in our design. (**a**) The three-dimensional (3-D) structure and simulation setup in HFSS. (**b**) The top view and relevant parameter notations.

**Figure 2 sensors-23-04264-f002:**
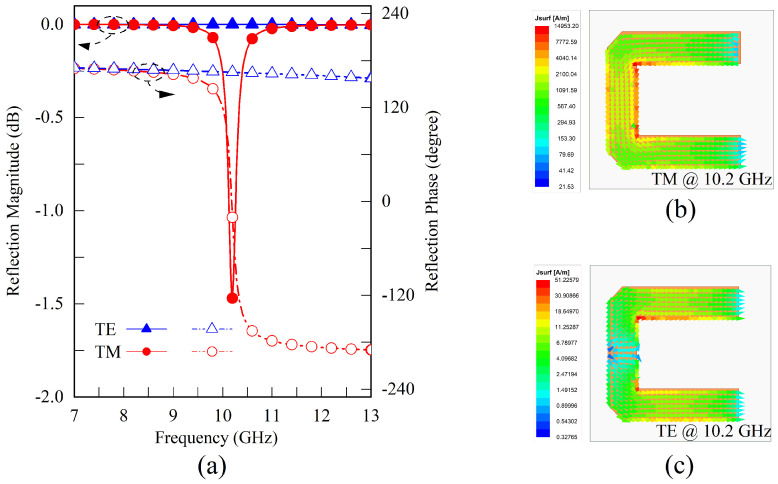
Simulation results of the U-shaped resonator without ground via under TE and TM polarizations. (**a**) The reflection magnitude and phase versus operating frequencies under different polarizations. (**b**) The TM mode current distribution of the U-shaped metallic surface at 10.2 GHz. (**c**) The TE mode current distribution of the U-shaped metallic surface at 10.2 GHz.

**Figure 3 sensors-23-04264-f003:**
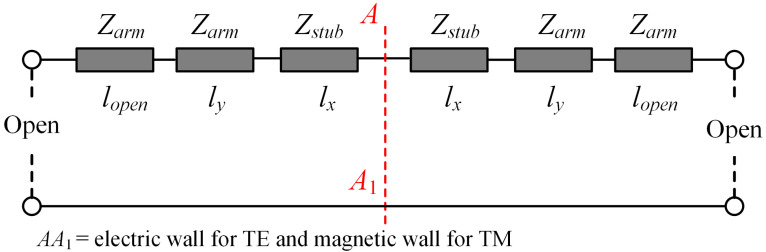
Equivalent transmission line model of the U-shaped resonator without grounded via.

**Figure 4 sensors-23-04264-f004:**
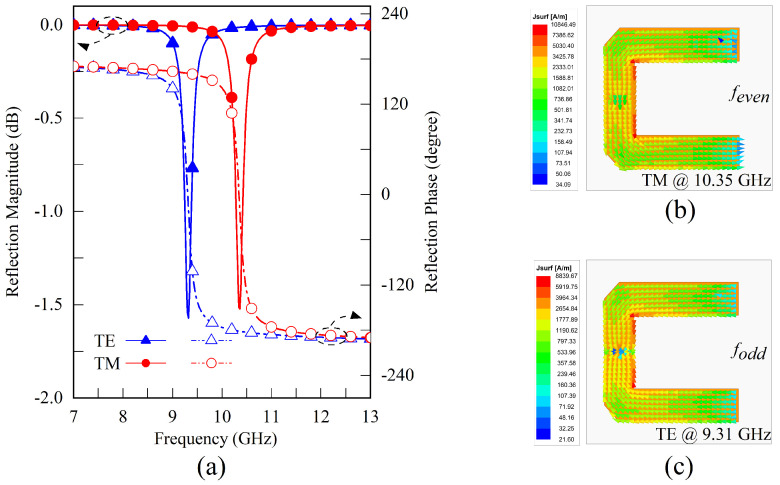
Simulation results of the U-shaped resonator with one ground via under TE and TM polarizations. (**a**) The reflection magnitude and phase versus operating frequencies under different polarizations. (**b**) The TM current distribution, which provides an even mode resonance at the frequency of 10.35 GHz. (**c**) The TE current distribution provides an odd mode resonance at a resonating frequency of 9.31 GHz.

**Figure 5 sensors-23-04264-f005:**
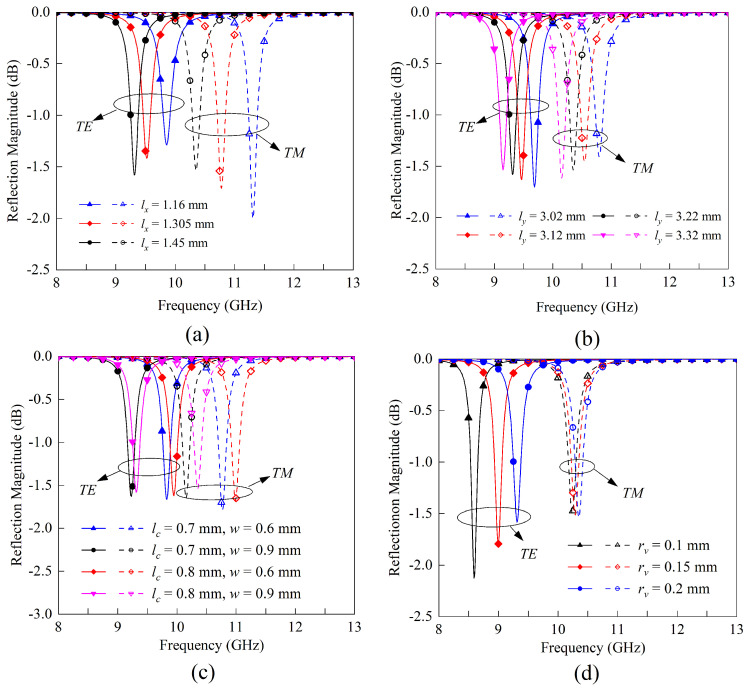
Full-wave simulation validation of the parametric study. (**a**) Resonance frequency tuning by adjusting the stub length $l_{x}$. (**b**) Resonance frequency tuning by adjusting the arm length $l_{y}$. (**c**) Resonance frequency ratio tuning by adjusting the width of the stub and arms. (**d**) TE resonance frequency tuning by adjusting the via radium.

**Figure 6 sensors-23-04264-f006:**
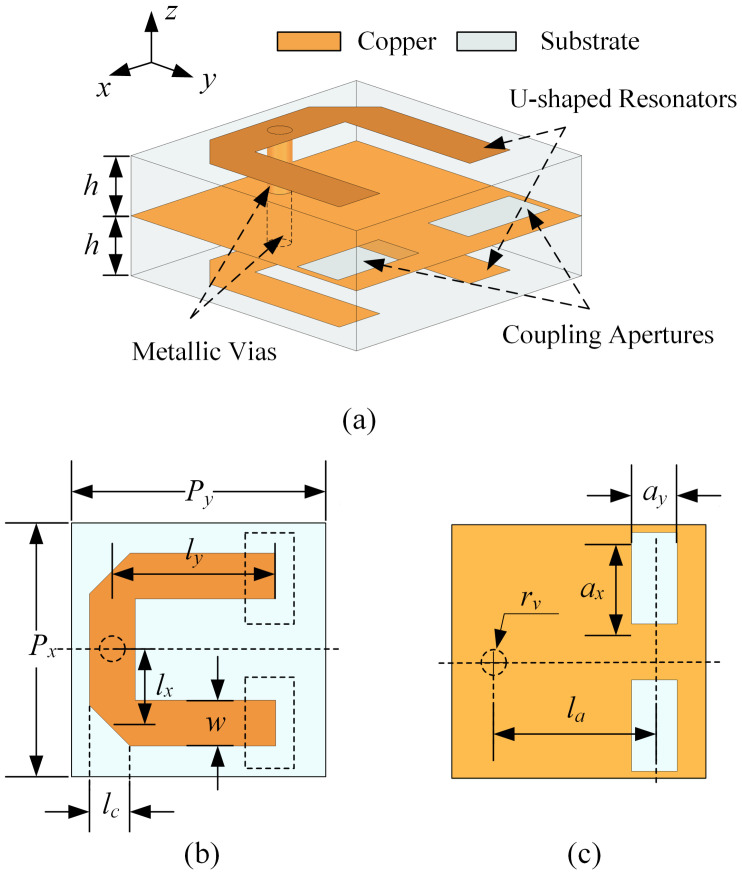
The proposed FSS unit cell. (**a**) The three−dimensional structure. (**b**) The top layer. (**c**) The ground plane with two etched apertures.

**Figure 7 sensors-23-04264-f007:**
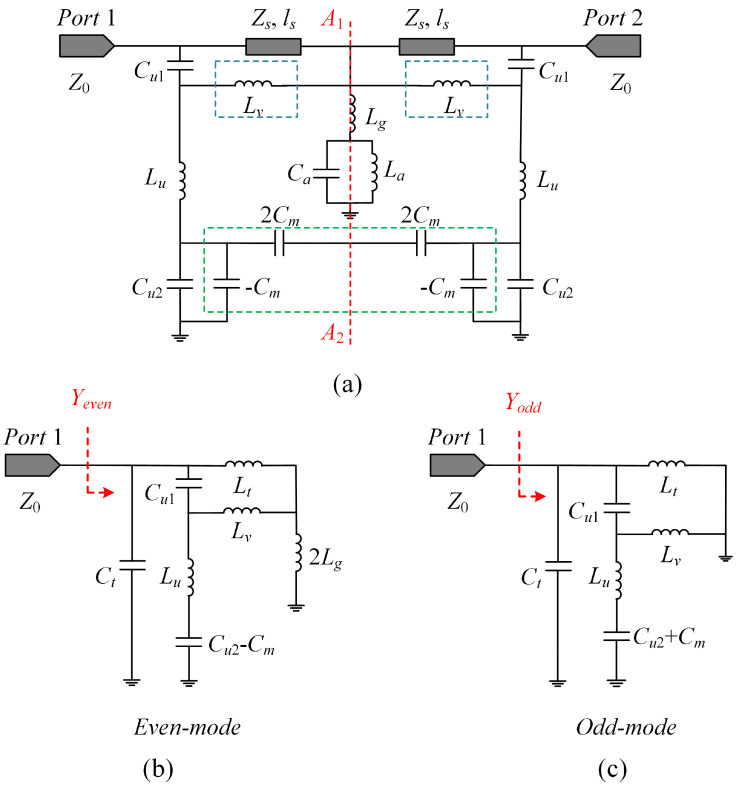
The analysis of the equivalent circuit model of the proposed FSS (**a**) The equivalent circuit model of the whole FSS. (**b**) The even mode equivalent circuit. (**c**) The odd mode equivalent circuit.

**Figure 8 sensors-23-04264-f008:**
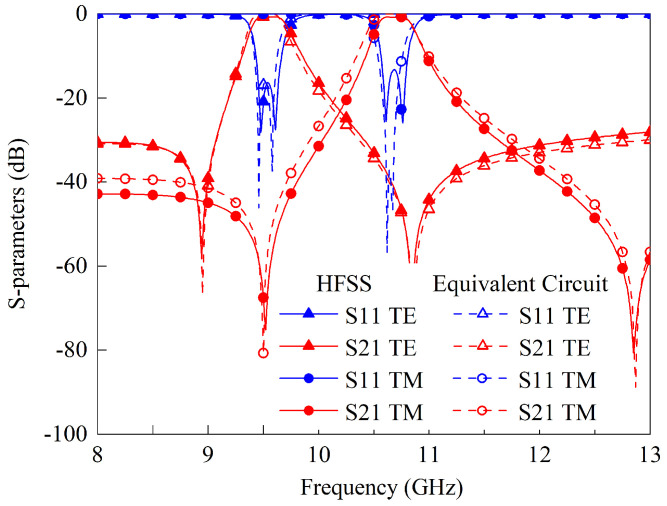
The S-parameters obtained by equivalent circuit analysis and full-wave simulation.

**Figure 9 sensors-23-04264-f009:**
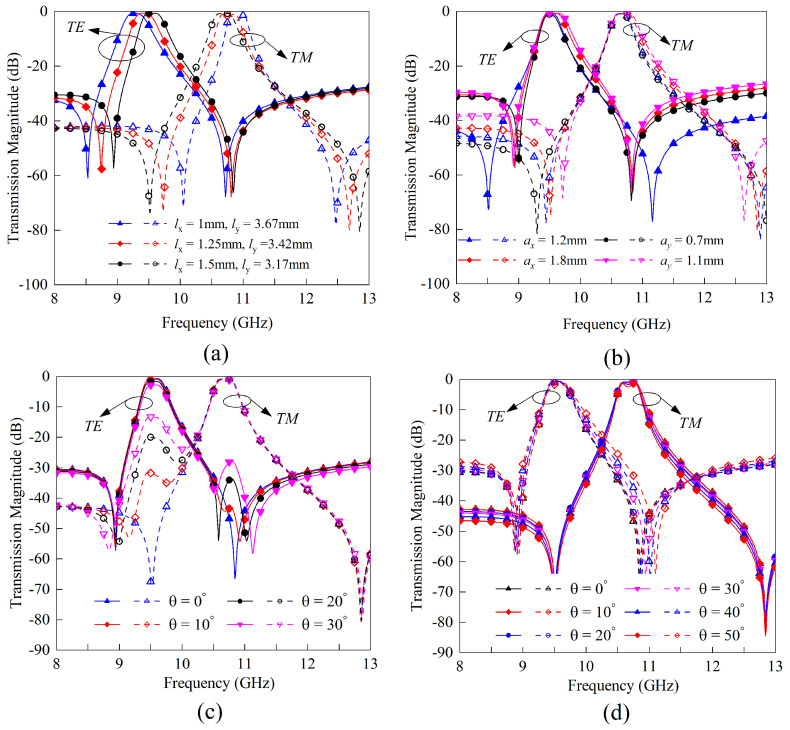
The effect of the geometric parameters and incident angle on FSS transmission magnitudes. (**a**) Different $l_{x}$ while the sum of $l_{x}$ and $l_{y}$ is fixed. (**b**) Different $a_{x}$ and $a_{y}$. (**c**) Different incident angles in xoz plane. (**d**) Different incident angles in yoz plane.

**Figure 10 sensors-23-04264-f010:**
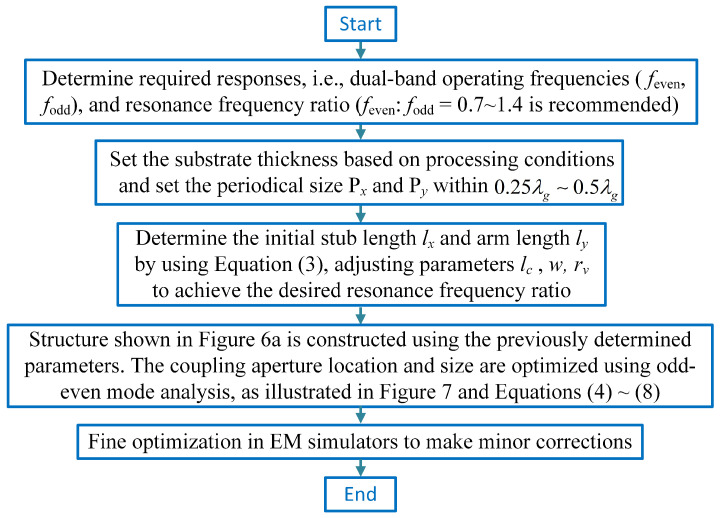
General flowchart of the proposed frequency− and polarization−selective surfaces design procedure.

**Figure 11 sensors-23-04264-f011:**
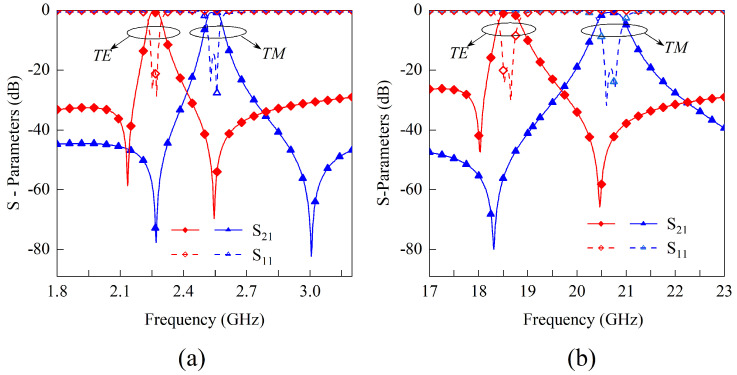
Designing the examples realized by utilizing the methods in Figure 10. The desired in−band minimum polarization isolation is 40 dB. (**a**) S−band design for indoor communication with center frequencies located at 2.35 and 2.55 GHz. (**b**) K−band design with center frequencies located at 18.7 and 21.6 GHz.

**Figure 12 sensors-23-04264-f012:**
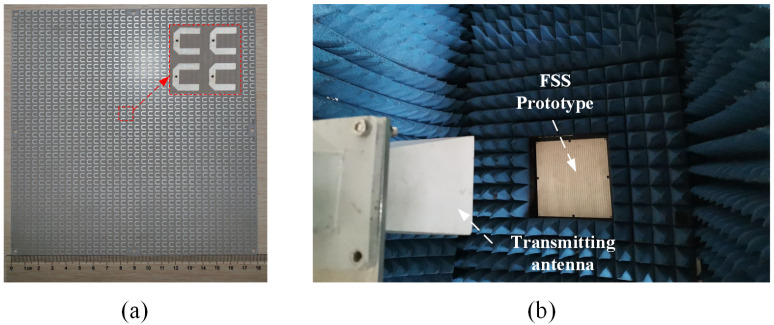
The measurement of the fabricated X-band design. (**a**) The FSS prototype with the dimension listed in Table 2. (**b**) The measurement setup.

**Figure 13 sensors-23-04264-f013:**
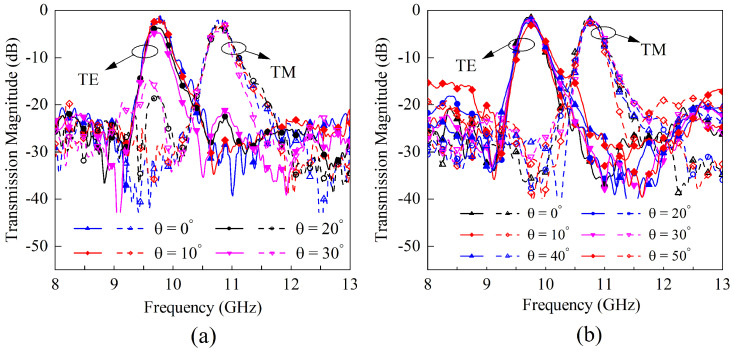
The measured transmission magnitude results under different incident angles. (**a**) in the xoz plane. (**b**) in the yoz plane.

**Table 1 sensors-23-04264-t001:** Geometric parameters of the adopted U-shaped unit cell.

Parameter	$l_{x}$	$l_{y}$	$l_{c}$	$P_{x}$	$P_{y}$	*h*	*w*	$r_{v}$
Value (mm)	1.45	3.22	0.8	5.0	5.0	0.508	0.9	0.2

**Table 2 sensors-23-04264-t002:** Geometric parameters of the double layers FSS.

Parameter	$l_{x}$	$l_{y}$	$l_{c}$	$P_{x}$	$P_{y}$	*w*	$r_{v}$	$l_{a}$	$a_{x}$	$a_{y}$
Value (mm)	1.45	3.22	0.8	5.0	5.0	0.9	0.2	3.19	1.8	0.9

**Table 3 sensors-23-04264-t003:**
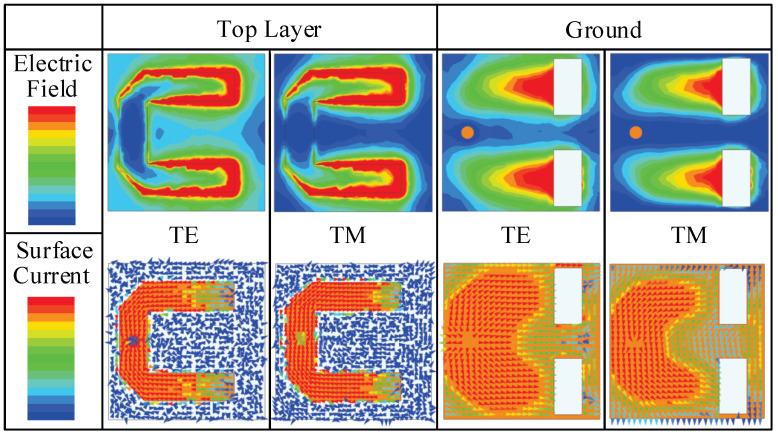
Electric field and surface current distributions of the U-shaped resonators and ground. The operating frequencies depicted in the screenshot are 9.54 GHz (TE) and 10.56 GHz (TM), respectively.

**Table 4 sensors-23-04264-t004:** Parameters of the equivalent circuit models.

TE Polarization	C/L ^1^	$C_{u1}$	$C_{u2}$	$C_{t}$	$C_{m}$	$L_{g}$	$L_{t}$	$L_{u}$	$L_{v}$
	Value (pF/nH)	0.093	0.511	0.008	0.38	0.125	0.638	8.91	3.44
TM Polarization	C/L	$C_{u1}$	$C_{u2}$	$C_{t}$	$C_{m}$	$L_{g}$	$L_{t}$	$L_{u}$	$L_{v}$
	Value (pF/nH)	0.061	0.170	0.008	0.05	0.028	0.638	8.04	3.44

^1^ C = Capacitance, L = Inductance.

**Table 5 sensors-23-04264-t005:** Geometric parameters of the two additional FSS design examples ^1^.

	C.F. ^2^	*h*	$l_{x}$	$l_{y}$	$l_{c}$	$P_{x}$	$P_{y}$	*w*	$r_{v}$	$l_{a}$	$a_{x}$	$a_{y}$
Figure 11a	2.25/2.55	2.5	6	4	3	25	25	4	1	6.5	8	4
Figure 11b	18.7/20.7	0.3	0.8	1.5	0.3	3	3	0.4	0.1	0.68	1.15	0.42

^1^ Operating frequency and physical dimensions are in GHz and mm. ^2^ C.F. = Center Frequency.

**Table 6 sensors-23-04264-t006:** Performance comparisons of the fabricated X−band FSS with other state of the art similar works.

Reference	Technology	C.F. (GHz)	FBW (%)	Worst IL ^1^ (dB)	Pol. Iso. ^2^ (dB)	Profile (mm, $\lambda_{0}$)	Order	TZ
[29]	Cylindrical dielectric resonators	7.4&11 (TE) 9.2 (TM)	17.5&14.5 (TE) 21 (TM)	0.9	30	4 $0.1\lambda_{0}$/$0.12\lambda_{0}$	2	2
[33]	Jerusalem cross	7.3&26.9 (TE) 11.4 (TM)	21.5&11.3 (TE) 26.1 (TM)	1.2	13.7	1.0∗2 $0.05\lambda_{0}$/$0.076\lambda_{0}$	2	0
[34]	Patch with twisted slot	8.2/13.58	16/4.5	2.4	14.2	1.5∗2 $0.08\lambda_{0}$/$0.14\lambda_{0}$	2	0
[46]	Patch resonator	3.34/4.41	9/4.5	2	19.5	3.15 $0.035\lambda_{0}$/$0.05\lambda_{0}$	2	0/2
[47]	Multi-mode patch	3.03/4.33	2.44/2.46 ^3^	1.06	12	3.1 $0.32\lambda_{0}$/$0.44\lambda_{0}$	2	1
[48]	Cross-dipole, loop	15/23 (Sim. ^4^)	10.7/11.7 (Sim.)	-	>12	1.03 $0.05\lambda_{0}$/$0.08\lambda_{0}$	2	4
[49]	Cross-dipole, loop	16/28 (Sim.)	35/45 (Sim.)	0.9∼1.4 (Sim.)	>20	7.5 $0.4\lambda_{0}$/$0.7\lambda_{0}$	2	0
This work	Grounded U-shaped resonator	9.68/10.73	3.45/3.48	1.1	30	1.07 $0.03\lambda_{0}$/$0.036\lambda_{0}$	2	2

^1^ Worst IL = The worst in-band insertion loss value. ^2^ Pol. Iso. = Polarization isolation under normal incidence wave. ^3^ Shape factor: the ratio between 10-dB and 1.5-dB bandwidth. ^4^ Sim. = Only simulation results provided.

## Data Availability

Not applicable.

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
