# Peer review of "Dual-Band Frequency Selective Surface with Different Polarization Selectivity for Wireless Communication Application"

_sensors, 2023, doi:10.3390/s23094264_

Round 1

Reviewer 1 Report

The manuscript presented a metasurface to support a dual-band FSS with two passbands operating at X-band. The manuscript is well written in general; find below some minor comments:

1.     In the last paragraph in the introduction section, the authors say that the general design flow diagram is the most important contribution to this manuscript. I guess you refer to figure 10. Since this is the main contribution, the authors have to provide a more detailed flowchart and validate, if possible, with equations the numerically suggested ranges of the recommended design parameters reported in the flowchart.

2.     What is the primary purpose of proving an equivalent circuit model (figure 7) for the proposed unit cell since the L’s and C’s are optimized to fit the result from the full wave simulation closely? Elaborate more and show the advantages of the provided equivalent circuit model.

Author Response

The authors would like to thank the editor and reviewers for their constructive comments and suggestions that have helped improve the quality of this manuscript. The manuscript has been thoroughly revised according to the editor's and reviewers’ comments. In this response letter, the authors' replies are shown in blue font, while the corresponding revisions in the revised manuscript are marked in red font. We hope that the quality of our manuscript has improved and that the revised version can be accepted by the reviewers and editors. A detailed response letter is attached here.

Reviewer 2 Report

1-The article proposed and investigated a dual-band, frequency- and polarization-selective surface. Multiple resonant modes are introduced using the U-shaped resonator with a ground via to achieve dual-band responses and polarization selectivity. Two symmetrically grounded U-shaped resonators are coupled through electrically coupled apertures in a common ground, resulting in a passband with two transmission zeros per polarization. 2-A general design flowchart and additional examples at the S, X, and K-bands are presented as well.  There is good analysis and extensive simulations, that are supplemented with experimental verifications.

3-In Page 2, it is mentioned that " TM wave reflectivity is approximately 40 dB at its highest". Can the authors explain how the reflected wave is larger than the incident one?.

4-In page 5, Line 151; the effective dielectric constant is assumed as (epsilon-r + 1)/2. why the other formula that considers the width of the strip and the substrate height was not used?.

Author Response

(The authors gave the same response as above.)

Reviewer 3 Report

This article proposes a dual-band, frequency- and polarization-selective surface using a U-shaped resonator with a ground via to achieve dual-band responses and polarization selectivity. The proposed designs have high selectivity, small frequency ratio, low profile, and stable performance under oblique incidence, making them potentially useful in wireless communication systems.

The contributions seem significant and valuable in the field of wireless communication and MIMO systems.

However, the authors should pay attention to and address the following points in the article.

1. Can you explain the methodology used to obtain the values presented in Table 2? Was any optimization technique utilized?

2. How were the parameter values of the equivalent circuit models, which are presented in Table 4, determined?

3. If optimization techniques were utilized, it is important to mention the range, default values, and algorithms that were employed.

4. Table 6 only compares the study with a total of 4 other studies. It would be beneficial to include more studies for comparison if they are available in the literature.

5. The conclusion section should be arranged to include and summarize the some of the numerical result.

Author Response

(The authors gave the same response as above.)
